# A Cross-Cultural Analysis of Plant Resources among Five Ethnic Groups in the Western Himalayan Region of Jammu and Kashmir

**DOI:** 10.3390/biology11040491

**Published:** 2022-03-23

**Authors:** Shiekh Marifatul Haq, Musheerul Hassan, Rainer W. Bussmann, Eduardo Soares Calixto, Inayat Ur Rahman, Shazia Sakhi, Farhana Ijaz, Abeer Hashem, Al-Bandari Fahad Al-Arjani, Khalid F. Almutairi, Elsayed Fathi Abd_Allah, Muhammad Abdul Aziz, Niaz Ali

**Affiliations:** 1Department of Botany, University of Kashmir, Srinagar 190006, India; marifat.edu.17@gmail.com; 2Wildlife Crime Control Division, Wildlife Trust of India, Noida 201301, India; 3Islamia College of Science and Commerce, Srinagar 190002, India; musheer123ni@gmail.com; 4Department of Ethnobotany, Institute of Botany, Ilia State University, Tbilisi 0162, Georgia; rainer.bussmann@iliauni.edu.ge; 5Institute of Food and Agricultural Sciences, University of Florida, Gainesville, FL 32611, USA; calixtos.edu@gmail.com; 6Department of Botany, Hazara University, Mansehra 21300, Pakistan; fbotany@yahoo.com (F.I.); niazalitk25@gmail.com (N.A.); 7Center of Plant Sciences and Biodiversity, University of Swat, Swat 19200, Pakistan; shaziasakhi@gmail.com; 8Botany and Microbiology Department, College of Science, King Saud University, Riyadh 11451, Saudi Arabia; habeer@ksu.edu.sa (A.H.); aalarjani@ksu.edu.sa (A.-B.F.A.-A.); 9Department of Plant Production, College of Food and Agriculture Science, King Saud University, Riyadh 11451, Saudi Arabia; almutairik@ksu.edu.sa (K.F.A.); eabdallah@ksu.edu.sa (E.F.A.); 10Department of Environmental Sciences, Informatics and Statistics, Ca’ Foscari University of Venice, Via Torino 155, 30172 Venezia, Italy; m.aziz@studenti.unisg.it

**Keywords:** conservation, ethnic groups, Venn diagram, ethno-usage pattern, Western Himalayas

## Abstract

**Simple Summary:**

For generations, local ethnic communities have amassed a vast body of traditional ecological knowledge (TEK) on the use of plant resources. Ethnobiologists have recently focused on cross-cultural studies in order to record and measure the processes guiding the evolution of TEK within a specific society; both to preserve it and use it sustainably in the future. The current study documents the TEK of plant resources from five ethnic communities of the Jammu and Kashmir (J&K) region, Western Himalayas. Through semi-structured interviews and group discussions, we recorded a total of 127 plant species used by local ethnic groups for various provisioning services (i.e., medicine, food, fodder, fuelwood, herbal tea) and/or with spiritual significance. Across the ethnic groups, Gujjar reported the highest number of plants (25% species), followed by Pahari (24% species), and the lowest number of plants were reported by Dogra (12% species). Looking at plant uses among different cultural groups, we discovered that, especially, some ritual practices were associated with specific plants. We found a relatively high overlap in the use of specific plants among the ethnic groups, namely Gujjar, Bakarwal, and Pahari. Certain species were found to be common in all cultures due to their food value. The current study is a collaborative effort that includes not only documenting, but also cross-cultural comparisons of the documented species, in order to better understand the diverse traditional plant usage systems. This will not only increase regional understanding of cross-cultural ethnobotany, but it will also open opportunities for local people to be rewarded for promoting and celebrating their expertise and participating in future development activities.

**Abstract:**

Plant resources have always been valuable in human life, and many plant species are used in medicine, food, and ritual, and resource utilization is closely related to cultural diversity. Our study was conducted from June 2019 to April 2021, during which we aimed to document the local knowledge of plant resources of five ethnic groups, i.e., the Gujjar, Bakarwal, Kashmiri, Pahari, and Dogra communities of the Jammu and Kashmir (J&K) region, Western Himalayas. Through semi-structured interviews (N = 342) and group discussions (N = 38), we collected data on the ethnobotanical uses of plant resources. The data was subjected to hierarchical cluster analysis and ordination techniques (Principal Component Analysis) via, R software of version 4.0.0. Traditional uses were classified into three groups, i.e., single-, double-, and multi-use groups. The study recorded a total of 127 plant species, belonging to 113 genera and distributed among 64 botanical families. The dominant plant families were the Asteraceae, with 8% of all species, followed by Lamiaceae (6%), Polygonaceae (5%) and Ranunculaceae (4%). The recorded plant taxa were frequently used for medicine (51.4% responses), followed by food (14.9%), and fodder (9.5%). Principal component analysis (PCA) separated three groups of provisioning services depending on plant consumption preference levels. Comparative analysis showed remarkable similarities in plant uses (food, medicinal) among the Gujjar and Bakarwal ethnic groups, as both groups share a common culture. Some plants like *Azadirachta indica, Brassica campestris, Ulmus wallichiana, Amaranthus blitum*, and *Celtis australis* were also used for magico-religious purposes. We also recorded some medicinal uses that are new to the ethnobotanical literature of the J&K Himalayas, such as for *Betula utilis, Sambucus wightiana*, and *Dolomiaea macrocephala*, in our case for example local medicinal recipe, which is derived from *Dolomiaea macrocephala,* often known as *Nashasta*, used to treat weakness, back pain, and joint pain. Similarly, we also recorded new food uses for *Eremurus himalaicus*. Moreover, we also observed some plants for instance, *Fragaria nubicola, Betula utilis* and *Juniperus communis* have spiritual significance (i.e., amulets and scrolls) for this part of the Himalayan region. The present study provides a useful tool for resource management and can help in developing scientifically informed strategies for the conservation of plant resources.

## 1. Introduction

The Himalayan Mountain region is home to a wide diversity of medicinal and food plants and is considered an important site of bio-cultural diversity - in other words, the site of *biocultural refugia* [1]. To date, a significant number of ethnobotanical studies have been carried out in different countries across this mountain range, including the Indian Kashmir, and have reported an enormous number of aromatic, medicinal, and food plants, being used by the local communities [2]. Particularly the tribal communities of the region are more dependent on non-timber forest products, and they extract their livelihood from plant-derived ingredients which are playing a crucial role in managing the traditional health care systems [3,4].

According to various studies, the Indian Himalaya is home to over 8000 species of vascular plants, with 1748 of them being known for their therapeutic potential [5,6]. Plant use is important for supporting the lives of tribal peoples living in the Himalayas by providing both food and medicine [2,3]. Since ancient times, many wild and cultivated plants have been used as curative agents, and medicinal plants have recently acquired popularity as natural ingredients for cosmetics as well as herbal medications [7]. Similarly, wild food plants (WFPs) are also important food ingredients, supporting the local food systems among various ethnic communities who are living in remote mountain territories. Reports have shown that medicinal plants which grow in the wild are frequently used by certain nomadic groups who are living in remote mountain areas and are using these plants to treat livestock diseases [8]. Recent research has also recorded cross-cultural uses of plant resources e.g., in the Balti, Beda, and Brokpa communities in the Ladakh region of the Trans-Himalayas. By offering an in-depth understanding of the ethnomedicinal, cultural, and ritual perspectives of plant diversity in the Ladakh region, this study investigated how the wild flora of Ladakh could benefit local life and aid poverty alleviation [4]. It has been affirmed that many plant species have been reported across the region and the local communities have developed an enormous body of traditional knowledge on the utilization of these natural resources since generations [9]. Because of rapid globalization and modernization, traditional ecological knowledge is a subject of discussion among contemporary ethno-biologists. Studies have shown that the remarkable social change has led Traditional Ecological Knowledge (TEK) to fade, if not erode [10]. On the one hand, we see the remarkable socio-cultural integration in the globalized world but on the other hand we are losing culture and languages. The three diversities of life i.e., biodiversity, cultural diversity, and linguistic diversity are facing equal threats. The socio-cultural integration of ethnic groups has caused cultural and linguistic homogenization, which in turn has also have homogenized the TEK [11,12]. Traditional knowledge is particular to each individual group, and it is vulnerable to change on a spatio-temporal basis, especially when held by small groups. In recent times, ethnobiologists have been focusing on cross-cultural studies to record and measure the dynamics leading to the evolution of TEK within a given society and put recommendations for policy frameworks. According to the recommendation given by the Convention on Biological Diversity [13], local knowledge should be part of future developmental processes to get sustainability because sustainability could be a praxis without considering the local wisdom- referred to as TEK of communities that have a longstanding relationship with their natural resources including plants. As Maffi et al. [14] described, a holistic approach should be adopted to counter the upcoming extinction crisis to ensure the sustainability of the planet. In this overwhelming situation, researchers need to focus on local and traditional knowledge to preserve it as a basis for future sustainability. We argue that the field of ethnobiological studies will not only help to protect TEK but will also encourage policymakers to focus on the social sustainability of ethnic groups to achieve the future long-term sustainable goal. The current study is not just a documentation but also a cross-cultural comparison of the documented taxa to understand various traditional plant use systems, highlighting the historical stratifications and economic status of the research groups. This will not only advance the knowledge of cross-cultural ethnobotany in the region, but it will also open potential lines for obtaining incentives for the local people to promote and celebrate their knowledge and get benefits in future development programs. This study focused on the comprehensive assessment of plant resources with the following objectives: (1) to document the ethnobotanical uses of the local flora among the different ethnic groups in the Jammu and Kashmir Region, (2) cross-cultural comparison of the ethnobotanical uses of the quoted plants, (3) to evaluate the conservation status of plant resources utilized to put policy recommendations for policymakers and relevant stakeholders. 

## 2. Materials and Methods

### 2.1. Study Area and Communities

The study area (Jammu and Kashmir) is part of Western Himalaya and currently administered as a Union Territory in India (Figure 1). The region is situated to the West of Ladakh UT, North of Himachal Pradesh, West of Punjab, and shares the international boundaries with Pakistan and China (Tibet) to the East. J&K has two biogeographic provinces, i.e., Jammu and Kashmir. Geo-graphically, Jammu and Kashmir comprise rugged mountains and barren slopes with Dfb (Warm-summer humid continental climate) climate category according to Koppen classification [15]. The main Himalayan range runs along the valley’s northeastern flank. The Kashmir valley has an average elevation of 1850 m above sea level (masl), but the surrounding Pir Panjal range has an elevation of 3000 masl. The Jammu region has an average elevation of 300 masl. The mighty mountain ranges support unique forest ecosystems with diverse microclimates [16]. The biodiversity and vegetation heterogeneity of the region is controlled by a combination of various biophysical, geographical, and topographical features [17]. Jammu and Kashmir are gifted with rich flora diversity with enormous economic potential. *Trillium govanianum, Fritillaria cirrhosa, Aucklandia costus, Aconitum heterophyllum, Dolomiaea macrocephala, Bergenia ciliata*, and *Rheum webbianum* were important medicinal plants collected by the indigenous population for their livelihood. The region is providing the home for linguistic communities like Kashmiri, Gujjar, Pahari, Dogra, and Bakarwal. The Kashmiri are the descendants of an Indo-European ethno-linguistic group [18], the Pahari trace their decent from the Kash empire [19], the Dogra trace their identity from the Ikshvaku (Solar) dynasty of northern Northern India [20], and the Gujjar and Bakarwal are believed to have migrated from Gujrat (via Rajasthan) and the Hazara Division of the Northwestern Frontier province [21]. According to the last census (2020) (https://uidai.gov.in/images/state-wise-aadhaar-saturation.pdf, accessed on 14 May 2020), the population of Jammu and Kashmir is 13606320. The sex ratio is 889 women per 1000 men. The inhabitants in the regions follow different faiths; 68.3% of the people are Muslim, 28.4% are Hindu, 1.9% Sikh, 0.9% Buddhist, and 0.3% Christian. 

### 2.2. Field Study

A field ethnobotanical study was carried out from June 2019 to April 2021 in 76 villages (Figure 1) in Jammu and Kashmir India. Information was gathered from several linguistic and religious groups distributed throughout the valley’s settlements. Semi-skilled workers, shepherds, farmers, daily wage laborers, govt. employees, housewives, skilled workers, shopkeepers, and others were recruited among middle-aged, young, and elderly residents (range: 18–75 years old), using a snowball strategy (Table 1). Of all informants, 76% were men and 24% were women. The reason that our study had less women participating lies in the social, religious, and cultural values of the society. Women, overall, are much more restricted to the household, and often not allowed or not willing to interact freely with strangers. Participants gave verbal prior informed consent before each interview, and the International Society of Ethnobiology’s Code of Ethics (ISE, 2008) was observed. Semi-structured interviews (N = 342) and group discussions (N = 38) were con-ducted in the local languages, (one of the authors was familiar with all the languages) [4,22]. The interviews focused on uses of the local flora for medicinal, religious, cultural, veterinary, and food usage. The survey results were redisplayed to the informants after data collection to remove errors and omissions from the data. Based on the uses of plant resources, traditional uses were classified into three groups i.e., single, double, and multi-use groups [23]. The conservation assessment of plant species was assessed as per International Union for Conservation of Nature (IUCN) criteria (version 3.1) [24]. Collected plant specimens were identified with the help of taxonomists at the CBT Lab, University of Kashmir, Srinagar (Jammu and Kashmir), and verified at the Herbarium of Kashmir University (KASH), where all vouchers were deposited. The POWO (Online flora) 2019 (www.plantsoftheworldonline.org, accessed on 26 October 2019) database was used for the authentication of the botanical names of the plants. We also compared our data with previous field ethnobotanical studies carried out in the nearby regions [2,4,6,25,26].
biology-11-00491-t001_Table 1Table 1Demographic status of the respondents from Jammu and Kashmir, Western Himalayas, India.Demographic FeaturesTotalBiogeographic Provinces(Linguistic) Ethnic GroupsJammuKashmirKashmiriPahariBakarwalGujjarDograRespondents48923325619481716974Language
Kashmiri  UrduPahari  UrduGujari  UrduGujari  UrduDogri  UrduGender
Male37216121114871445158Female11762554610271816Age range (years)
Young (18–26)10143573716181614Middle (27–55)13661744922241823Old (56–75+)25210215210843293537Profession
Farmers20812112223Shepherds5622141492175Semi-skilled Workers11343708215088Daily wage laborers82344841210713Govt. employees321022105557Housewives542529101115108Skilled workers733142111226159Shopkeepers59283115621521Livelihood source


Horticulture  &  Cattle rearingHorticulture  &  Cattle rearingPastoralismAgriculture PastoralismSericulture  Agriculture  Cattle rearingPopulation 


51.72%7.80%11.90%9.05%20.04%Descendants/Migrated from


Indo-EuropeanKash empireMigrated from GujratMigrated from Northwestern Frontier ProvinceIkshvaku (Solar) dynasty

### 2.3. Use Value (UV)

The UV is used to determine the relative value of a species in relation to other species [27], and is calculated as:UV = ΣU/N(1)
where U = number of use reports for a given species, and N = total number of informants. A high UV score suggests that there are numerous usage reports for that plant, whereas a low score shows that the informants provided fewer use reports for that plant.

### 2.4. Data Analysis

To evaluate the difference in the number of plant parts used, we used a Generalized Linear Model (GLM) with binomial distribution, followed by a Likelihood Ratio test using the packages “stats” [28] and “car” [29]. We used a heat map associated with a cluster analysis using the Sorensen’s similarity index [30] to elucidate species grouping based on presence–absence data [31]. Using this method, similar groups cluster close to each other, and dissimilar groups are shown as distant clusters. Principal component analysis (PCA) was employed to visualize the provisioning services and plant parts associations between plants using the package “*vegan*” [32] in the software R ver. 4.0.0 [28]. The contribution of different usages was displayed in chord diagrams using the “circlize” package [33] in R software 3.6.1 [28]. The chord diagram allows us to investigate which plant species are associated with which use groups, as well as count the number of species in each use group and determine which use group is more diverse based on the thickness of each bar [34]. The Venn diagram was created to see the variation in the use of plant resources among the five ethnic groups, using the Bioinformatics & Evolutionary Genomics Webpage [35]. 

## 3. Results

### 3.1. Plant Composition and Distribution

This study documented a total of 127 plant and fungal species belonging to 113 genera in 64 families (Table 2). Even though this number may appear low, especially in light of the area’s high biodiversity [36,37], the species number of useful plants identified was comparable to studies published by other researchers from various Himalayan regions. For instance, Mustafa et al. [38] reported 121 plant species from Azad Kashmir, Pakistan while Kayani et al. [39] reported 125 plant species from mountainous forests of Pakistan with ethno-botanical relevance. Similarly, in the Naran Valley of Western Himalayas of Pakistan, Khan et al. [40] reported 101 medicinal plant species. Angmo et al. [41] reported 160 plant species from Western Ladakh. Awan et al. [42] reported ethnobotanical knowledge of 102 medicinal and aromatic plants from Mansehra Dis-trict, Pakistan. Similarly, Singh et al. [43] from western Himalayas-India reported 78 plant spe-cies with ethnomedicinal attributions. Raj et al. [44] from Northern Bengal-India reported 140 ethnomedicinal plants.

### 3.2. Species Distribution in Families

Across the 64 families, the distribution of species was unequal, and half of the species be-longed to just 13 families (Asteraceae, Lamiaceae, Polygonaceae, Ranunculaceae, Fabaceae, Pina-ceae, Plantaginaceae, Rosaceae, Rutaceae, Apiaceae, Berberidaceae, Boraginaceae, and Malva-ceae) while the remaining half belonged to 51 families. A large number of families (38) were monotypic i.e., with one species in each (Table 1), similar to other studies [45,46,47]. The domi-nant plant families included Asteraceae with 10 species (8%) followed by Lamiaceae (7 species, 6%), Polygonaceae (6 species, 5%) and Ranunculaceae (5 species, 4%). The details of reported plant species are given in Table 2. The prevalence of the aforementioned families was comparable to studies from other Himalayan regions [2,48,49]. The members of the Asteraceae family accli-mate quickly and adapt to arid settings due to their wide ecological amplitude [50]. In contrast, Angmo et al. [41] reported Polygonaceae as the dominant family from Cold Desert Biosphere Re-serve, India, and Kayani et al. [39] described Ranunculaceae as the most dominant representative family in mountain forests of Pakistan, and Hussain et al. [51] reported Lamiaceae as the domi-nant family from the Koh-e-Safaid Range of Pakistan. Despite their diversity, members of these families are distinguished by their ability to synthesize secondary metabolites with potentially significant biological activity. As a result, they are used in a variety of ways in traditional medi-cine to treat a wide range of ailments [52].

### 3.3. Plant Usage Pattern

Based on the uses of plant resources, traditional uses were classified into three groups i.e., single-, double-, and multi-use groups (Figure 2).

Single usage: Plants used for a single purpose, e.g., *Aconitum chasmanthum*, *Acorus calamus*, *Ajuga parviflora*, *Arnebia benthamii*, *Corydalis govaniana*, *Dactylorhiza hatagirea*, *Euphorbia wallichii*. The majority (52%) of the plants in the study area had a single usage.

Double usage: Plants used for two applications, e.g., *Achillea millefolium*, *Berberis asiatica*, *Bergenia ciliata*, *Fragaria nubicola*, *Indigofera heterantha*, *Justicia adhatoda*, *Pinus wallichiana*, *Rheum webbianum*, *Rhododendron arboreum* and *Taxus wallichiana*. Around 32% of the plants in the study area had double usage.

Multi usage: Plant species such as *Abies pindrow*, *Acer caesium*, *Aesculus indica*, *Betula utilis*, *Cedrus deodara*, *Malva neglecta*, and *Parrotiopsis jacquemontiana* were used for more than two purposes and classified as multi-usage plants. Around 16% of the plant species reported in the study area had multiple-usage.

The use value (UV) indicates the importance of a species to the informants and the local ethnomedicinal system. The main species (Table 2) in the current study with the highest UVs are *Dolomiaea macrocephala* (UV = 0.28), *Aconitum heterophyllum* (0.27), *Artemisia absinthium* (0.29), *Podophyllum hexandrum* (0.27). Whereas the lowest UV value is recorded for *Albizia lebbeck* and *Sambucus wightiana* (0.11). The high UV of medicinal plant species in the study region is attributed to their common usage of plants across ethnic groups in the area, and the local people are well familiar with their medicinal uses. Plants were mostly harvested for self-consumption (55%) and for commercial purposes (45%). The highest priority of the local people was the extraction of medicinal plants in the summer season (79%), followed by the early autumn season (21%). The indigenous population collected important medicinal plants (especially *Trillium govanianum, Fritillaria cirrhosa, Aucklandia costus, Aconitum heterophyllum, Dolomiaea macrocephala, Bergenia ciliata*, and *Rheum webbianum*) for both traditional use and trade. The region’s medicinal plant trade is the single most important factor in the preservation of traditional knowledge about medicinal plants. The continued use of medicinal plants demonstrates the value of these ancient practices [53]. 

The leaves of *Aesculus indica*, *Celtis australis*, *Betula utilis*, *Parrotiopsis jacquemontiana*, *Indigofera heterantha*, *Salix alba*, *Acer caesium*, *Ziziphus jujuba*, and *Prunus cornuta* were used as fodder. The plant species like *Heracleum candicans*, *Trifolium pratense*, *Trifolium repens*, and *Taraxacum officinale* were used to increase the milk production in cattle. *Achillea millefolium* and *Artemisia absinthium* were fed to animals to remove worms. The fresh leaves of *Cannabis sativa* were crushed and applied topically to kill lice and ticks. A decoction made from the roots of *Bergenia stracheyi* was used as an antiseptic treatment of foot and mouth disease. The mashed roots of *Rheum webbianum* were used to cure indigestion and constipation in cattle, and the paste of its roots and leaves was used to treat external injuries. The sap of the leaves and stem from *Sambucus wightiana* was applied to wounds in cattle. *Malva neglecta* was used against weakness. Jan et al. [54] also reported that, because of the therapeutic properties of plants, ethnic communities use them to treat a variety of animal ailments, and they play an important role in the rural veterinary healthcare system. Similarly, Rafique Khan et al. [55] reported 39 plant species used by the indigenous communities of Kashmir Himalaya for curing 21 different diseases of 7 different types of livestock. Abassi et al. [56] reported a total of 89 species of 46 families used for ethnoveterinary applications.
biology-11-00491-t002_Table 2Table 2List of plant species recorded and their ethno-botanical usage in Jammu and Kashmir, Western Himalaya, India.FamilyScientific Name  Voucher Number  (Abbreviation)Local NameParts UsedPreparationsEthnobotanical ApplicationsConservation StatusEthnic GroupsUse ValueCitationAdoxaceae*Sambucus wightiana* Wall. ex Wight &Arn.SMH-495(Sam.wig)Ganullo (G)Faqual (B)Fallo (K)Felo (P)Leaves Roots StemFruitFresh roots Leaves are dried and powderedRoots are eaten in small amounts to treat asthma. (G,B)Sap of leaves and stem are applied for wound healing, mostly in cattle. (G,P)Mature fruits are used as an alternative as adhesive locally. (K) Least ConcernGujjar (N = 15)Bakarwal (N = 13)Kashmiri (N = 16)Pahari (N = 12)0.1156Agaricaceae*Bovista plumbea* Pers.SMH-426(Bov-plu)Madaam (G)Mangri (B)Sore (P)Fruiting bodyRaw fruiting body.Fruiting bodies are used as food and as homeopath medicine. *Least ConcernGujjar (N = 19)Bakarwal (N = 22)Pahari (N = 24)0.1365*Scleroderma bovista* Fr.SMH-500(Scl-bov)Mongar (G)Mangd (P)Mangdi (B)Fruiting bodyFresh fruiting body Fruit body is cooked. *Least ConcernGujjar (N = 18)Bakarwal (N = 22)Pahari (N = 24)0.1364Amaranthaceae*Amaranthus blitum* L.SMH-414(Ama-blit)Kanhaar (G)Ganhar (k)Gan (D)Rata (P)SeedsLeavesSeeds are dried and powdered.Leaves are boiled, fried with spices.Seeds are used to cure continuous sneezing and rhinorrhea, back pain. (G,P) Leaves used as food. (K, D)Seeds are also used to wind off the ill effects of black magic. (K, D)Least ConcernGujjar (N = 25)Kashmiri (N = 29)Pahari (N = 23)Dogra (N = 20)0.2097*Celosia argentea* L.SMH-429(Cel-arg)Mawal (G)Mawl (D)SeedsLeavesSeeds are dried and powdered.Seeds with water are used for anti-diarrheal and leaves are eaten as vegetables. *Least ConcernGujjar (N = 55)Dogra (N = 50)0.21105Amaryllidaceae*Allium semenovii* RegelSMH-413Wan pran (G)Jangli pran (B)Prun (P)Roots LeavesCooked.Roots and leaves are used as vegetables, spices and condiments. *Least ConcernGujjar (N = 30Bakarwal (N = 32)Pahari (N = 27)0.1889Apiaceae*Angelica glauca* Edgew.SMH-416(Ang-gla)Choro (G)Chre (P)Sore (B)Leaves RootsCooked.A recipe is made from the leaves in combination with the kidney bean curry used to treat obesity (G,P) Roots are used as spice. *EndangeredGujjar (N = 31)Bakarwal (N = 47)Pahari (N = 45)0.25123*Berula erecta* Huds… SMH-424(Ber-ere)Coville (G)Shungji (P)RootsRoots are eaten raw.Roots are eaten and held in the mouth to relieve toothache. *Least ConcernGujjar (N = 32)Pahari (N = 43)0.1575*Bunium persicum* Boiss.(SMH-308)Zuur (K)Zeera (B,P,G)Kala zeera (D)SeedRawCookedSeeds are cooked with rice locally called as *zeera rice.*
Gujjar (N = 19)Bakarwal (N = 24)Kashmiri (N = 39)Pahari (N = 21)Dogra (N = 15)0.24118*Heracleum candicans* Wall. ex DCSMH-454(Her-can)Wanntamokh (G)Tamboko (P)Tamooq (B)RootsLeavesRoots are made into paste.Leaves dried.Root paste is applied on joints by arthritis patients to relieve pain. (G) Leaves are used in cigarettes. (P) Fresh aerial parts eaten by shepherds as salad. * Leaves are locally given to cattle to increase milk production. (G,P)Least ConcernGujjar (N = 19)Bakarwal (N = 23)Pahari (N = 32)0.1574Apocynaceae*Calotropis procera* (Aiton) W.T. AitonSMH-513(Cal-pro)Aak (G)Aaak (D)FruitFlowerRaw fruit is used.Fruit is used as poison (G).In Hinduism flowers are used to praise their deity *Hunuman. **Least ConcernGujjar (N = 14)Dogra (N = 68)0.1782Asphodelaceae*Eremurus himalaicus* BakerSMH-446(Ere-him)SheilHaakh (G)Haaq (B)LeavesStemCookedYoung leaves are used as vegetables. (G)Stem is chewed and used in raw form for juice (B)Least ConcernGujjar (N = 38)Bakarwal (N = 41)0.1679Asteraceae*Achillea millefolium* LSMH-403(Ach-mil)Gandhna (G) Dare (B)Pahelgassa (K)Kaa (P)Root Leaves StemCrushed.Roots are used against toothache. (K) Aerial parts are used as fodder and crushed into balls then fed animals to remove abdominal worms. (B,P,G)Least ConcernGujjar (N = 22)Bakarwal (N = 28)Kashmiri (N = 34)Pahari (N = 28)0.23112*Artemisia absinthium* LSMH-419(Art-abs)Tethyan (K)Chawoo (G)Chaw (P)Leaves FlowersLeaves and flowers are dried and powdered and taken with water. Leaves and flowers used as anthelmintic and used to treat joint pain. *Flowers are fed to animals to remove abdominal worms. (K,P)Least ConcernKashmiri (N = 68)Gujjar (N = 38)Pahari (N = 36)0.29142*Aucklandia costus* Falc.SMH-496(Auc-cos)Kuth (K)Kushtha (P,B)Kosath (D)RootsLeavesRoots are used fresh and also sun dried.Leaves are cooked.Roots are used for treatment of dysentery, rheumatism, skin disorder, cough, cold and bronchial asthma. * Leaves are used as vegetables. (P,D)Critically EndangeredBakarwal (N = 26)Kashmiri (N = 32)Pahari (N = 38)Dogra (N = 14)0.23110*Cichorium intybus* L.SMH-532(Cic-int)Posh hand (K)Saze hand (P)Padee (G)RootsLeavesLeaves and roots are cooked.Syrup is made from roots.Leaves are given as vegetables to women during pregnancy. (K) Roots are used to fight typhoid. (G) Whole plant is used to treat ulcers and as a blood purifier. (P) Root is cooked as a vegetable. *Least ConcernGujjar (N = 36)Kashmiri (N = 51)Pahari (N = 41)0.26128*Dolomiaea macrocephala* DC. ex RoyleSMH-462(Dol-mac)Guggal Dhoop (G,P)Thandijaid (K)RootsRaw roots are usedRoots are used as a stimulant and treatment for fever and back pain. (G,P)The roots are also used for making traditional medicine recipes locally called *Nashasta.* (K)Used to treat body weakness and lower back pain. *Least ConcernGujjar (N = 43)Kashmiri (N = 64)Pahari (N = 22)0.28129*Himalaiella heteromalla* (D. Don) Raab-StraubeSMH-497(Him-het)Kalizri (G)Kalzre (P)LeavesRootsLeaves are made into paste and mixed with mustard oil.Roots are made into decoction.Leaf paste with mustard oil is massaged on leucoderma and wounds. *Roots are used to treat fever. (G)Least ConcernGujjar (N = 41)Pahari (N = 52)0.1993*Inula royleana* DCSMH-458(Inu-roy)Poshkar (G)Pushkarmula (B)Hasubkual (P)Maleen (P)RootsDried roots are used.Roots are used as disinfectant, mainly used to protect garments from insect damage. *Least ConcernGujjar (N = 21)Bakarwal (N = 23)Pahari (N = 34)0.1678*Saussurea amabilis* Kitam.SMH-498(Sau-ama)Brahmkamal (G,P)Pangchi (B)LeavesDecoction is obtained from leaves.Leaves are used for treatment of paralysis, wounds, pain and urinary problems. *Critically EndangeredGujjar (N = 32)Bakarwal (N = 19)Pahari (N = 47)0.2098*Saussurea roylei* (DC.) Sch. BipSMH-499(Sau-roy)Koth (G)Qoath (B)Oath (P)Whole plantPlant is shade dried and powdered.Dried powdered is used in the treatment of wounds, excessive bleeding, and meat poisoning. *Least ConcernGujjar (N = 34)Bakarwal (N = 19)Pahari (N = 47)0.21100*Taraxacum officinale* F.H. WiggSMH-504(Tar-off)Hand (K)Handri (G,P)Haandi (B)Karti (D)Leaves RootsLeaves are boiled and fried with spices.Leaves are also boiled in water for more than an hour.Roots are boiled in water.Leaves are used as vegetables and roots are also edible, especially given to the ladies who have given birth to babies. * Leaves are also used with other herbs for bathing the same ladies and their young ones. (K)Roots are used as fodder to increase milk production. (G,P)Least ConcernGujjar (N = 19)Bakarwal (N = 24)Kashmiri (N = 39)Pahari (N = 21)Dogra (N = 15)0.24118Araceae*Acorus calamus* L.SMH-407(Aco-cal)Bariyan (G)Bareen (P)Vai Vai (B) Gander (K)RhizomesRhizomes are dried and powdered.Rhizomes are used for treating digestive and nervous disorders. *Least ConcernGujjar (N = 28)Pahari (N = 32)Bakarwal (N = 19)Kashmiri (N = 59)0.28138Balsaminaceae*Impatiens glandulifera* RoyleSMH-456(Imp-gla)Treeli phal (G,P)Masar (B)Flowers Whole plantFlowers boiled in water for 5 min.Plants are made into paste.Flowers are used as a cooling tonic. (G,B) Plant paste is applied to joints to relieve pain. (P) Seeds are eaten raw. *Least ConcernGujjar (N = 33)Bakarwal (N = 38)Pahari (N = 43)0.23114Berberidaceae*Podophyllum hexandrum* RoyleSMH-481(Pod-hex)Banwagun (G)Gulkakkri (P)Papri (B)Soz (K)RootsFruitsRoots are dried, powdered and taken with water.Roots are used for the treatment of lung cancer. (K) Fruits are eaten when ripe. *Critically EndangeredGujjar (N = 35)Kashmiri (N = 37)Bakarwal (N = 28)Pahari (N = 32)0.27132*Berberis asiatica* Roxb. ex DCSMH-420(Ber-asi)Kaimbli (P)Kingora (G)Dandleder (K)RootsStem FruitsDried roots are boiled to get black liquid.Roots used as laxative and tonic. (P,G) Stem is used as fuel wood. * Fruits are eaten by children. *Least ConcernPahari (N = 42)Gujjar (N = 25)Kashmiri (N = 32)0.2199*Berberis lycium* RoyleSMH-421(Ber-lyc)Kamblu (G)Sumblu (G)Kawdasch (K)Chonphal (P)RootsStem FruitsRoots are dried, powdered and taken with water.Roots are used for bleeding piles and fever. (K) Stem is used as fuel wood. *Fruits are eaten by children. *Least ConcernGujjar (N = 17)Kashmiri (N = 52)Pahari (N = 48)0.24117Betulaceae*Betula utilis* D. DonSMH-425(Bet-uti)Bhojpatra (G)Burza (K)Borzal (B)Burez (P)Burjaa (D)BarkStem leavesBark is boiled in water for more than 30 min.Bark is used to make tea by the locals and also used by spiritual healers to write scrolls. * Stem is used as fuelwood. (P,G) Stem is cut and modified into glass like vessel in which water is put overnight take empty stomach in morning to treat diabetes. (K) Leaves are used as fodder. *EndangeredGujjar (N = 21)Bakarwal (N = 15)Kashmiri (N = 23)Pahari (N = 21)Dogra (N = 15)0.1995Boraginaceae*Cynoglossum glochidiatum* Wall. ex Benth.)SMH-435(Cyn-glo)Cherun (G)Chree (P)SeedsSeeds kept in water for some time before use.Seeds are used against erectile dysfunction and for improvement of fertility. *Least ConcernGujjar (N = 36)Pahari (N = 42)0.1678*Arnebia benthamii* (Wall. ex G. Don) I.M. JohnstSMH-417(Arn-ben)Kahzabaan (K,B,G,P)Gaozabaan (D)RootsFlowersRoots are dried, powdered and used with lukewarm water.Flowers are dried and powdered.Dried powdered roots with lukewarm water are used against fever and cough. * Flowers are used as cardiac drug. (G) Meanwhile tea is also obtained from both roots and flowers (K,P)Critically EndangeredGujjar (N = 29)Bakarwal (N = 21) Kashmiri (N = 36)Pahari (N = 21)Dogra (N = 27)0.27134*Arnebia euchroma* RoyleSMH-418(Arn-euc)Zabermuks (G) Raktmundi (B)Jadi (P)Ratanjog (K)Rentnigog (D)RootRoot extract mixed with butter.Root is used for treating hair growth problems *.Critically EndangeredGujjar (N = 20)Bakarwal (N = 19)Kashmiri (N = 23)Pahari (N = 24)Dogra (N = 16)0.21105Brassicaceae*Capsella bursa-pastoris* (L.) *Medik*SMH-427(Cap-bur-pas)Kralmond (K)Soontsabz (B)Qoralmunj (P,G)Karl (D)LeavesShootsRaw leaves and shoots are used.Leaves and shoots are taken with meals as vegetables, also used as fodder. *Least ConcernGujjar (N = 15)Bakarwal (N = 29)Kashmiri (N = 32)Pahari (N = 20)Dogra (N = 18)0.26114*Brassica campestris* LSMH-523(Bra-cam)Tilgagul (K)Sarsoon (G,P)Sarcoo (D)SeedsSeed oilSeeds are dried, powdered and taken with milk. Dried seeds are taken to a local oil extracting machine to get oil.Seeds are used to improve impotence. (K)Seed oil is used in cultural festivals like Diwali to light the lamps called Deepak. * Seeds are also to wind off evil. *Least ConcernGujjar (N = 20)Kashmiri (N = 31)Pahari (N = 29)Dogra (N = 28)0.22108Cannabaceae*Celtis australis* LSMH-430(Cel-aus)Bramij (K)Batkal (P)LeavesStemFruitsFresh leaves, dried stems and raw fresh fruits are used.Leaves used as fodder and stem as fuelwood. * Fruits eaten raw by children. *Small stem is hung in the house for the protection of children from evil eyes. *Least ConcernKashmiri (N = 84)Pahari (N = 19)0.21103*Cannabis sativa* L.SMH-517(Can-sat)Bhang (G)Bang (P)Pang (B, D)Charas (K)FruitsLeavesFruits are made into paste. Fresh leaves are crushed and applied topically.Fruits are used for the treatment of psoriasis, itching and leprosy. * Paste of leaves is served with drinks in the festival “Holi” called Bhang. *Least ConcernGujjar (N = 20)Bakarwal (N = 05)Kashmiri (N = 14)Pahari (N = 13)Dogra (N = 21)0.1573Caprifoliaceae*Dipsacus inermis* WallichSMH-442(Dip-ine)Shinglin (P) Mingli (P)LeavesDecoction is obtained from leaves.Leaves are used to cure swellings and pain. * Leaves are also given to cows after delivery to keep healthy and increase milk yield. *Least ConcernPahari (N = 63)0.1363*Morina coulteriana* RoyleSH-468(Mor-cou)Jamnoo (G)Bhuss (P)RootsRoots are shade dried.Roots are used to protect garments from insect damage. *Least ConcernGujjar (N = 21)Pahari (N = 38)0.1959Cupressaceae*Juniperus communis* LSMH-460(Jum-com)Yathur (B)Bitru (G)Bita (P)Whole plantPlant is burned to ashes.Plant (Ash) is used for tooth aching. Above ground parts are used as fuel wood by nomadic people, leaves are used as alternatives to incense. *Least ConcernBakarwal (N = 26)Gujjar (N = 21)Pahari (N = 38)0.1785*Juniperus squamata* Buch.-Ham. ex D. DonSMH-461(Jun-squ)Yathur (B)Bita (P)Butul (G)Whole plantPlant is sun dried and powdered and mixed with water.Plant is used in different skin diseases. (G,P) Above ground parts are used as fuel wood. (P) Leaves are used as an alternative to incense. *Least ConcernBakarwal (N = 24)Gujjar (N = 21)Pahari (N = 27)0.1572Dioscoreaceae*Dioscorea belophylla* (Prain) Voigt ex HainesSMH-440(Dio-bel)Tilpush (G)Tilmoha (G)Contres (D)RhizomesCookedRaw rhizomes are cooked, used for treatment of piles, dysentery, cough and cold. *Least ConcernGujjar (N = 41)Dogra (N = 38)0.1679*Dioscorea deltoidea* Wall. ex GrisebSMH-441(Dio-del)Tard (K)Kinns (G)Krees (P)Descoria (P)Aerial partRhizomesAerial parts are cooked.Raw rhizomes are used.Aerial parts are consumed as vegetables. * Rhizomes are used for treatment of digestive and abdominal disorders. (G,K)Critically EndangeredGujjar (N = 22)Kashmiri (N = 41)Pahari (N = 40)0.21103Dryopteridaceae*Dryopteris stewartii* Fraser-Jenk.SMH-443(Dry-ste)Kunji (G)Daid (K)Dedui (P)Young frondsCookedYoung fronds are used as vegetables. *Least ConcernGujjar (N = 35)Kashmiri (N = 41)Pahari (N = 39)0.24115Ephedraceae*Ephedra gerardiana* Wallich ex C. A. MeyerSMH-444(Eph-ger)Sutuchur (G)Chhepath (G)Asmanibuti (P)Young frondsFruitsRaw young fronds are usedYoung fronds are used for treatment for bronchitis, cold, cough and asthma. * Fruits are sometimes eaten raw. *VulnerableGujjar (N = 65)Pahari (N = 69)0.27134Equisetaceae*Equisetum arvense* L.SMH-445(Equ-arv)Gandumgud (K)Sategandie (K)Whole plantPlant is dried and powdered and then taken with cow milk.Plant is used for acidity, kidney infection and toothaches, meanwhile plants is also used for washing utensils *.Least ConcernKashmiri (N = 100)0.20100Ericaceae*Rhododendron arboreum* Sm.SMH-489(Rho-arb)Nichhni (G,B)Rattanbat (D)FlowersFlowers are boiled in water and taken orally.Flowers boiled in water are used for the treatment of headache, diabetes, rheumatism *.Least ConcernGujjar (N = 20)Bakarwal (N = 36)Dogra (N = 18)0.1574*Rhododendron campanulatum* D. DonSMH-490(Rho-cam)Cheu (G) Hardhulla (G)Madhal (B)Burans (B)Leaves FlowersRaw leaves are used.Flowers are sun dried.Leaves are used to induce vomiting. (B) Dried petals are used for making tea. * Flowers are used wishing the splendor of crops for never ending food security. *Least ConcernGujjar (N = 39)Bakarwal (N = 43)0.1782Elaeocarpaceae*Elaeocarpus angustifolius* BlumeSMH-515(Ela-ang)Rudraksh (G,D)SeedsDried seeds are used.Seeds are used to Worship Lord Shiva a deity in Hinduism. *Least ConcernDogra (N = 74)Gujjar (N = 06)0.1680Euphorbiaceae*Euphorbia wallichii* Hook. fSMH-448(Eup-wal)Konpal (G)Pencil (B)Hirbi (B)Stem sapStem is nailed and sap is obtained.Stem sap is used for treatment of rheumatism, neuralgia, toothache and against skin problems. *Least ConcernGujjar (N = 51)Bakarwal (N = 59)0.22110Fabaceae*Albizia lebbeck* (L.) BenthSMH-412(Alb-leb)Siris (G)Sareen (G)Sirinn (D)Shirish (D)BarkBark is dried, powdered and used with honey.Dried powdered bark is used in treating bronchitis. *Least ConcernGujjar (N = 31)Dogra (N = 24)0.1155*Dalbergia sissoo* DSMH-531(Dal-sis)Sheeshum (G)Shasm (B)Shai (D)StemLeavesDried stem is used.Stem is used as timber and leaves as fodder. *Least ConcernGujjar (N = 25)Bakarwal (N = 27)Dogra (N = 18)0.1470*Indigofera heterantha* BrandisSMH-437(Ind-het)Zind (G)Kes (K)Yti (P)LeavesTwigsCooked.Leaves are used as vegetable. (P) Flexible long twigs are used to make traditional fire pots (*Kangri*) for heating purposes in winter. (K) Leaves are used as fodder. *Least ConcernGujjar (N = 39)Kashmiri (N = 31)Pahari (N = 35)0.21105*Trifolium pratense* L.SMH-507(Tri-pra)Khanda posh (K)Khubpos (G)Gujjurposh (P)Uyti (D)LeavesflowerCooked.Leaves are used as vegetables. * Leaves are also used as fodder to increase milk yield. (K) Flowers are eaten raw. *Least ConcernGujjar (N = 27)Kashmiri (N = 45)Pahari (N = 19)Dogra (N = 18)0.22109*Trifolium repens* L.SMH-508(Tri-rep)Khanda (G)Posh (P)Putre (D)Fool (K)LeavesflowerLeaves are boiled in water and fried with spices.Fresh parts are eaten as salad and leaves are used as vegetables, meanwhile leaves are also used as fodder to increase milk yield. * Flowers are eaten raw. (K)Least ConcernGujjar (N = 28)Kashmiri (N = 39)Pahari (N = 31)Dogra (N = 10)0.22108Gentianaceae*Gentiana kurroo* RoyleSMH-452(Gen-kur)Tratmaan (G)Karu (G)Pashanbhed (P)LeavesJuice is extracted from the leaves by grinding then squeezing in a cotton cloth.Leaves are used as bitter tonic for improving appetite and gastric secretion. *Critically EndangeredGujjar (N = 41)Pahari (N = 72)0.23113Geraniaceae*Erodium cicutarium* (L.) L’Hér.SMH-447(Ero-cic)Gardyan (K)Whole plantWhole plant is grinded and mixed with water.Whole plant is used as a uterine sedative and skin infection. *Least ConcernKashmiri (N = 65)0.1365Hamamelidaceae*Parrotiopsis jacquemontiana* (Decne.) RehderSMH-471(Par-jac)Pooh (K)Kodi (P)Kedai (G)Hosi (B)StemTwigsLeavesDried stems and twigs are used. Leaves are used in both dry and green form.Stem is used for handling agricultural equipment; fuel wood and leaves are used as fodder. * Twigs of flexible long twigs are used to make traditional fire pots (*Kangri*) for heating purposes in winter. *Least ConcernGujjar (N = 29)Bakarwal (N = 25)Kashmiri (N = 39)Pahari (N = 26)0.24119Iridaceae*Iris nepalensis* Lawr.SMH-459(Iri-nep)Mazar mond (K)RhizomesRaw rhizomes are used along with decoction.Rhizomes are used against rodents in apple orchards and vegetable gardens, decoction is used for treatment of rheumatism. (K)Least ConcernKashmiri (N = 103)0.21103Lamiaceae*Ajuga parviflora* Benth.SMH-411(Aju-par)Khurbanti (B)Jaan e Adam (P)LeavesRaw dried leaves are used.Leaves are used for gastric problems in children and also used to cure mouth ulcers. *Least ConcernBakarwal (N = 70)Pahari (N = 68)0.28138*Mentha arvensis* L.SMH-467(Men-arv)Pudino (P)Fadna (K)LeavesRaw leaves are used.Leaves are used against indigestion and stomach inflammation. * Leaves added as spice and condiment. (K)Least ConcernKashmiri (N = 94)Pahari (N = 30)0.25124*Plectranthus amboinicus* (Lour.) Spreng.SMH-480(Ple-amb)Patta (D)Ajwain (G)LeavesCooked.Leaves are used as vegetables, and also used for making tea. *VulnerableGujjar (N = 45)Dogra (N = 34)0.1679*Prunella vulgaris* L.SMH-484(Pru-vul)Kalveuth (K)Aerial partFlowersAerial partand flowers are boiled in water and used.Flowers are used for rheumatism and piles, and also for headaches and common colds. (K)Flowers are also used in bathing the ladies who have given birth to young ones. (K)Flowers are mixed with the flowers of other species like *Cichorium intybus*, *Arnebia benthamii* to make herbal tea. (K)Aerial parts are boiled in water, used to wash the feet of COVID-19 patients for lowering the body fever. (K)Least ConcernKashmiri (N = 100)0.20100*Thymus linearis* Benth.SMH-506(Thy-lin)Ajwain (G)Jaind (K)Jayeen (P)Whole plantsPlant is dried and powdered, used with water.Plant is used to treat coughs, relieve digestive gas, and protect against hookworm. (K)Leaves are used as spice and condiment * Whole plant is used to wash and clean the pots used for milk. (G,P)Least ConcernGujjar (N = 40)Kashmiri (N = 31)Pahari (N = 4070.24118*Vitex negundo* L.SMH-513(Vit-neg)Bana (P)Shimula (D)LeavesStemLeaves are dried and powdered.Leaves are used for febrifuge diuretic. (P) Stem is used as fuel wood in rural areas. (P) Leaves are also used as fodder *Least ConcernDogra (N = 74)0.1574*Origanum vulgare* L.SMH-522(Ori-vul)Wan baber (K)Bail (D)LeavesLeaves are shade dried.Leaves are also boiled in water for more than one hour and consumed.Leaves are used for promoting menstruation flow. (K)Leaves are also used to for bathing the ladies who have given birth to the new ones. (D)Least ConcernKashmiri (N = 51)Dogra (N = 55)0.1995Liliaceae*Fritillaria cirrhosa* D. DonSMH-451(Fri-cir)Sheethkar (B) Ksheerkakoli (K)Pranik (K)FlowersFlowers are shade dried and used with warm water.Flower is used for treatment of asthma. *Critically EndangeredBakarwal (N = 40)Kashmiri (N = 68)0.22108Lauraceae*Cinnamomum camphora* (L.) J. Presl.SMH-521(Cin-cam)Kafoor (K,G,P)Whole plantDry plant is used.Plant is used for timber and camphor. (P,G)Camphor is also used in the Muslim faith for bathing the deceased person. *Least ConcernGujjar (N = 31)Pahari (N = 28)Kashmiri (N = 35)0.1994Lythraceae*Woodfordia fruticosa* (L.) KurzSMH-514(Woo-fru)Khukni (D)Joiu (B)FlowersFlowers are dried in the sun for more than 3 days.Flowers are used for herbal tea *.Least ConcernBakarwal (N = 55)Dogra (N = 46)0.20101Malvaceae*Lavatera cachemiriana* CambessSMH-464(Lav-cas)Saz posh (K) Junglisonchal (B)LeavesLeaves are dried in the sun by covering with a fine cloth and then powdered and used with water.Leaves are used as blood purifiers and to check amnesia. *Least ConcernBakarwal (N = 52)Kashmiri (N = 51)0.21103*Malva neglecta* Wall.SMH-465(Mal-neg)Jungalisoxal (B)GurSachal (G)Suchal (P)Sochaal (K)LeavesSeedsLeaves are cooked, and seeds are dried grinded to powder and used with water.Leaves are cooked as food and powdered seeds are used as protein and fat supplements. *Used against weakness in cattle. (K,P)Least ConcernGujjar (N = 30)Bakarwal (N = 23)Kashmiri (N = 29)Pahari (N = 26)0.22108*Althaea officinalis* L.SMH-520(Alth-off)Sazposh (G)Khurposh (P)Pati (D)LeavesFlowersFresh leaves are cooked, dried leaves are powdered and used with water. Flowers are shade dried.Leaves are cooked as food and dried powdered leaves with water are used as gargle to treat mouth and throat ulcers. * In Muslim faith flowers are kept in water used in bathing the corpuses to perform last religious rituals before burial. (G,P)Least ConcernGujjar (N = 45)Dogr (N = 31)Pahari (N = 61)0.28137Marsileaceae*Marsilea minuta* L.SMH-466(Mar-min)Paflu (K)LeavesRhizomePetioleLeaves, rhizome, and petiole are dried, powdered and used with lukewarm water.Leaves are used against cough, and bronchitis. (K)Petiole and rhizome are used against typhoid. (K)Least ConcernKashmiri (N = 68)0.1468Melanthiaceae*Trillium govanianum* Wall. ex D. DonSMH-509(Tri-gov)Tripater (P)Satgandi (B)Tulhakh (G)RhizomesLeavesRhizomesare used raw and leaves are boiled and fried with spices.Raw rhizomes are used against skin irritation. (G) Leaves are used as vegetables. *VulnerableGujjar (N = 23)Bakarwal (N = 12)Pahari (N = 34)0.1669Meliaceae*Azadirachta indica* A. JussSMH-516(Aza-ind)Neem (D, P)LeavesFresh leaves are used.Leaves are kept in wardrobe, used to protect the clothes from insect eating. (P) In Hindu faith leaves extracts are believed to heal gods, hence are also eaten raw. (D)Least ConcernPahari (N = 57)Dogra (N = 50)0.22107Myrtaceae*Syzygium cumini* (L.) SkeelsSMH-503(Syz-cum)Jamun (G)Dhalla (D)BarkFruitJuice is extracted from the bark by grinding then squeezing.Bark juice is used for the treatment of liver disease and cancer. (G) Fruits are eaten raw. *Least ConcernGujjar (N = 68)Dogra (N = 46)0.23114Ophioglossaceae*Ophioglossum reticulatum* L.SMH-470(Oph-ret)Chonchur (B)Cokloiu (K)Young frondsCooked.Young fronds are used as vegetables and salads. *Least ConcernBakarwal (N = 50)Kashmiri (N = 84)0.27134Orchidaceae*Dactylorhiza hatagirea* (D. Don) SoóSMH-436(Dac-hat)Salam Panja (G)Hathajari (P)RootsRaw roots are used.Roots are used as energy boosters, help in improving health, and are recommended for weak people. * Roots are also used as nerve tonic. (G)Critically EndangeredGujjar (N = 57)Pahari (N = 68)0.25125Orobanchaceae*Pedicularis siphonantha* D. DonSMH-472(Ped-sip)Singmarore (B)Phakchang (B)FlowersDecoction.Used to treat edema and urinary disorder. (B)Least ConcernBakarwal (N = 61)0.1261Papaveraceae*Corydalis govaniana* Wall.SMH-432(Cor-gov)Bhut Kesi (B)Bhutjata (P)Nakpo (P)RootsFlowersRoots are grinded and kept overnight then squeezed. Flowers are macerated in water for three days and then whole content is used.Roots extracts are used as a tonic, antiperiodic. * Flowers are used against headaches. (P)Least ConcernBakarwal (N = 69)Pahari (N = 63)0.27132Phytolaccaceae*Phytolacca acinosa* Roxb.SMH-473(Phy-aci)Chamchi- Pata (G)BrandHakh (B)Hapat makai (P)Seeds LeavesSeeds of the plant are boiled in water and then taken orally.Leaves are boiled and fried with spicesSeeds are used to treat typhoid fever (G).Leaves are used as vegetables. *Least ConcernGujjar (N = 29)Bakarwal (N = 33)Pahari (N = 43)0.21105Pinaceae*Abies pindrow* (Royle ex D. Don) RoyleSMH-401(Abi-pin)Baddul (K)Cheeda (G,P)Cheed (B)Yuldr (D)StemBranchesBarkDried stem is used. Fresh branching is used. Dried bark is boiled in water for more than 30 min.Stem is used as timber and branches are used for flooring/bedding in nomadic huts (G,B,P). Bark is used for herbal tea. *Least ConcernGujjar (N = 23)Bakarwal (N = 34)Kashmiri (N = 15)Pahari (N = 22)Dogra (N = 19)0.1995*Cedrus deodara* G. DonSMH-428(Ced-deo)Devdar (K)Deaar (G,P,B)Davdar (D)StemBranchesBoth stem and branches are used in dry form.Stem as timber and branches for fuel wood. *Least ConcernGujjar (N = 22)Bakarwal (N = 25Kashmiri (N = 23)Pahari (N = 25)Dogra (N = 18)0.23113*Picea smithiana* (Wall.) BoissSMH-474(Pic-smi)Bunder (B)Budul (K)Budal (P)StemBranchesDried stem and fresh branching are used.Stem is used as timber (K, P) Branches are used for flooring/bedding in nomadic huts. (B)Least ConcernBakarwal (N = 35)Kashmiri (N = 29)Pahari (N = 36)0.20100*Pinus roxburghii* Sarg.SMH-476(Pin-rox)Chir (D)Chegai (B)Cheraa (G)StemSeedsDried raw seeds and stems are used.Stem is used for furniture making, house building, and fuel wood. * Seeds are eaten raw. *Least ConcernGujjar (N = 44)Bakarwal (N = 39)Dogra (N = 45)0.26128*Pinus wallichiana* A.B. Jacks.SMH-477(Pin-wal)Kayerd (B) Yaed (P)StemBranchesLeavesLeaves are cooked.Dry stems and branches are used.Stem is used as timber and branches as fuel wood. * Resin is used for healing wounds. * Leaves eaten as vegetable. (P)Least ConcernBakarwal (N = 67)Pahari (N = 66)0.27133Plantaginaceae*Digitalis purpurea* L.SMH-439(Dig-pur)Wopalhaakh (K)Dadid (B)LeavesFresh or dried leaves are boiled in water.Leaves are used by women for bathing after delivery. * Used as cardio stimulants. (B)Least ConcernBakarwal (N = 38)Kashmiri (N = 46)0.1784*Picrorhiza kurroa* Royle ex Benth.SMH-475(Pic-kur)Hanglang (G)Kali (B)Heeng (P)Roots FlowersDecoction is obtained from roots and flowers.Decoction from roots is used in asthmatic disorders, fever, and blood purification. (G) Flowers are used for malaria, biliousness, dropsy. *Critically Endangered (Endangered)Gujjar (N = 35)Bakarwal (N = 22)Pahari (N = 48)0.21105*Plantago lanceolata* L.SMH-478(Pla-lan)Gul (K)Sabaz gul (G)Nuulgul (P)Neela (D)Cheekdei (B)SeedsYoung leavesSeeds are dried and powdered and taken with lukewarm water.Young leaves are boiled and fried with spices.Seeds are used for soothing effect to mucus membranes of the intestine. * Young leaves are used as vegetables and fodder. *Least ConcernGujjar (N = 21)Pahari (N = 25)Bakarwal (N = 35)Kashmiri (N = 19)Dogra (N = 12)0.23112*Plantago major* L.SMH-479(Pla-maj)Bud-gull (K)Sabze (B)Sarbenosh (P)Purfers (D)Kapdan (G)LeavesYoung leaves are boiled and cooked, meanwhile, paste is also made.Leaves (paste) are applied on wounds, which stimulates tissue growth. *Young leaves are used as vegetables and fodder. (B)Least ConcernGujjar (N = 21)Bakarwal (N = 24)Kashmiri (N = 15)Pahari (N = 28)Dogra (N = 16)0.21104Poaceae*Cynodon dactylon* (L.) Pers.SMH-434(Cyn-dac)Khabbal (P)Dramn (K)Nekgi (D)RootsJuice is obtained by grinding and then squeezing.Drinking the juice with an empty stomach in the morning is good for normalizing sugar level (hyperglycemia). *Least ConcernKashmiri (N = 22)Pahari (N = 38)Dogra (N = 35)0.1995*Stipa sibirica* (L.) Lam.SMH-502(Sti-sib)Gud Gass (G)Ka (P)LeavesFresh leaves are usedUsed as fodder, flooring/bedding hut. *Least ConcernGujjar (N = 31)Pahari (N = 44)0.1575Polygonaceae*Fagopyrum esculentum* MoenchSMH-449(Fag-esc)Ogla (D)Kotu (K)RootsLeavesLeaves are cooked and decoction is made from roots.Root decoction is used for rheumatic pains, lung problems (D). Leaves are used as vegetables. *Least ConcernKashmiri (N = 47)Dogra (N = 38)0.1785*Polygonum amplexicaule* D. DonSMH-482(Pol-amp)Adder (G)Maachran (B)Chai (P)Rhizomes StemRhizomes and stem are dried and boiled in water for more than 20 min., meanwhile both are cooked also.Underground rhizomes, leafy stems are used for making tea and vegetables. *Least ConcernGujjar (N = 39)Bakarwal (N = 31)Pahari (N = 38)0.22108*Rheum moorcroftianum* RoyleSMH-487(Rhe-moo)Archa (G)Lachuu (B)Revandchini (P)Saiy (K)RootsRoots are dried and powdered and taken with lukewarm water.Roots are used in mild constipation, stomach problems and muscular swellings. *Critically Endangered (Vulnerable)Gujjar (N = 21)Bakarwal (N = 26)Kashmiri (N = 39)Pahari (N = 27)0.23113*Rheum webbianum* RoyleSMH-488(Rhe-web)Pambhak (K) Pambchalan (G,P)Pumush (B)LeavesRootsLeaves are dried and cooked.Roots are shade dried and powdered, mixed with water to form paste.Leaves are used as vegetables. * Root’s paste is applied to joints to get against pain. *VulnerableGujjar (N = 32)Bakarwal (N = 25)Kashmiri (N = 31)Pahari (N = 35)0.25123*Rumex dentatus* LSMH-491(Rum-den)Abij (K)Abjee (P,G)Parsee (B)LeavesCooked.Leaves are used as vegetables. *Least ConcernGujjarBakarwalKashmiriPahari0.1995*Rumex nepalensis* SprengSMH-492(Rum-nep)Junglipalak (K)Khembeer (B)Kushf (G)Kotre (P)LeavesLeaves boiled and fried with spices.Leaves are used as vegetables. *Least ConcernGujjarBakarwalKashmiriPahari0.21104Primulaceae*Primula denticulata* Sm.SMH-483(Pri-den)Kalashdandi (B)Qulsu (K)LeavesLeaves are sun dried powdered and taken with water.Dried powdered leaves are used to cure headache and liver ailments. *Least ConcernBakarwalKashmiri0.1678Pteridaceae*Adiantum capillus-veneris* L.SMH-409(Adi-cap)Hanspadi (G,P)Tryee (B)Hand (K)Dumtuli (D)Whole plantWhole plant ground to powder and taken with warm water.Whole plant is used as diuretic, also used to get against cough and skin disease. *Young leaves are also used as food. (P)Least ConcernGujjar (N = 23)Bakarwal (N = 21)Kashmiri (N = 15)Pahari (N = 19)Dogra (N = 18)0.1996Pyronemataceae*Geopora arenicola* (Lév.) Kers.SMH-453(Geo-are)Shajkan (K)Papdeyaan (P,G)Papdan (B)Fruiting bodyCookedWhole fruit body is used as a vegetable. *Least ConcernGujjar (N = 20)Bakarwal (N = 12)Kashmiri (N = 28)Pahari (N = 29)0.1889Ranunculaceae*Aconitum chasmanthum* Stapf ex HolmesSMH-404(Aco-cha)Atis (B)Ponkar (P)RootsRoots are dried grinded used with water.Roots dried, grinded, used for the treatment of pulsating headaches. *Critically EndangeredBakarwal (N = 68)Pahari (N = 41)0.22109*Aconitum heterophyllum* Wall. ex RoyleSMH-405(Aco-het)Patrees (B)Atees (G)RootsLeavesDried powdered roots are used with water.Leaves are cooked.Roots with water are used to treat diarrhea and dysentery. * Leaves used as vegetables. (G)Critically EndangeredBakarwal (N = 70)Gujjar (N = 63)0.27133*Aconitum violaceum* Jacquem. ex StapfSMH-406(Aco-vio)Dudhia (B)Atees (G)Dudhi mohra (G)LeavesCookedLeaves are used as vegetables. *VulnerableBakarwal (N = 66)Gujjar (N = 63)0.26129*Actaea spicata* L.SMH-408(Act-spi)Bhilar (B)RootsDecoctionDecoction is used as nervine sedative emetic and purgative. *Least concernBakarwal (N = 71)0.1771*Anemone tschernjaewii* RegelSMH-415(Ane-tsc)Rattanjogh (G)Belhar (B)Boi (P)RhizomeRhizome is grinded and taken with goat milk.Rhizome is used for treatment of acidity and joint pain. *Least concernGujjar (N = 30)Bakarwal (N = 21)Pahari (N = 44)0.1995Rhamnaceae*Ziziphus jujube* Mill.SMH-516(Ziz-juj)Ber (G)Beree (B)Singli (D)Fruits LeavesFruits are dried in the sun for more than a week.Dried fruits are eaten raw and used to treat irritability, insomnia, and heart palpitations. * Leaves are used as fodder. *Least concernGujjar (N = 36)Bakarwal (N = 31)Dogra (N = 32)0.2099Rosaceae*Crataegus songarica* K. KochSMH-433(Cra-son)Ring kol (K)Reng (P)FruitsLeavesFruits and leaves are taken as raw.Fruits (berries) are used for cardiac insufficiency. (K)Raw leaves are used as antioxidants. * Least concernKashmiri (N = 77)Pahari (N = 17)0.1994*Fragaria nubicola* Lindl. ex Hook. f.SMH-450(Fra-nub)Lacaita (B) Yangtaesh (G)Ingresh, (K)Rengresh (K)Ishtabur (P)FruitsLeavesFruits are taken as raw, also squeezed through a fine mesh to obtain ink.Fruits are eaten raw to improve digestion, anemia, treat tongue blemish and profuse menstruation. (G) The sap of fruits is locally used as ink by spiritual healers to write scrolls and amulets. * Leaves are used as fodder. *Least concernGujjar (N = 25)Bakarwal (N = 19)Kashmiri (N = 21)Pahari (N = 24)0.1889*Prunus cornuta* (Wall. ex Royle) SteudSMH-485(Pru-cor)Chuli (G,P,B)Cabt (K)Leaves StemLeaves are used fresh and stem in dry form.Leaves are used as fodder and stem as fuel wood. *Least ConcernGujjar (N = 21)Bakarwal (N = 18)Kashmiri (N = 31)Pahari (N = 38)0.22108*Pyrus pashia* Buch.-Ham. ex D. DonSMH-486(Pyr-pas)Kainth (G,B)Batangi (D)Fruits Leaves StemFruits and leaves are used raw. Stem is used in dry form.Fruits used for conjunctivitis and diarrhea. *Leaves are used as fodder and stem as fuel wood. (G)Least ConcernGujjar (N = 41)Bakarwal (N = 37)Dogra (N = 33)0.21111Rutaceae*Murraya koenigii* (L.) Spreng.SMH-469(Mur-koe)Drainkru (G)Gandhla (B)Goi (D)Tender leavesRaw tender leaves are used.Tender leaves are eaten in diarrhea and dysentery. (G)And young leaves are used as food. *Least ConcernGujjar (N = 29)Bakarwal (N = 34)Dogra (N = 32)0.1995*Skimmia laureola* Franch.SMH-501(Ski-lau)Butputer (G,P)Naer (B)Whole plantDecoction is made from leaves. Bark is dried and powdered.Bark is used for the healing of burns and wounds by applying topically. * Decoction of leaves are used for the treatment of headache. (G)Least ConcernGujjar (N = 29)Bakarwal (N = 38)Pahari (N = 42)0.22109*Aegle marmelos* (L.) CorrêaSMH-514(Aeg-mar)Bel patra (D)Bail pata (G)FruitsLeavesEaten raw.Fresh and dried fruits are eaten. *Leaves are used to praise the deity *Lord Shiva* in Hinduism. *Least ConcernGujjar (N = 54)Dogra (N = 31)0.1785*Zanthoxylum armatum* DCSMH-515(Zan-arm)Maratch (B)Mesrgh (D)Mreci (G)Whole plantFruits, seeds, and bark are sun dried and taken with water.Fruits, seeds, and bark are used as aromatic tonic in dyspepsia and fever. *Least ConcernGujjar (N = 38)Bakarwal (N = 41)Dogra (N = 35)0.23114Santalaceae*Santalum album* L.SMH-518(San-alb)Sandali (P)Chanadan (D)BranchesFresh pieces are used.Small pieces of branches are kept in houses in pockets for fragrance. * In Hinduism *Santalum album* is known to be sacred. (P)VulnerableDogra (N = 55)Pahari (N = 58)0.23113Salicaceae*Salix denticulate* AnderssonSMH-494(Sal-den)Jungaliyeed (K)Chede (P,G)Roster (D)Yeeri (B)Twigs Leaves StemFresh leaves and twigs are used. Stems are used when dry.A brush made of twig is rubbed gently around the teeth and gums to relieve toothache. (K) Leaves used as fodder and stems as fuel wood. *Least ConcernGujjar (N = 21)Bakarwal (N = 25)Kashmiri (N = 33)Pahari (N = 17)Dogra (N = 28)0.25124*Salix alba* L.SMH-493(Sal-alb)Beenso (G)Yeed (K)Yeer (B)Yekdi (P)Pasti (D)Twigs Leaves StemFresh leaves and twigs are used. Stems are used when dry.Leaves are boiled in water for more than one hour.A brush made of twig is rubbed gently around the teeth and gums to relieve toothache and also for tooth cleaning. * Leaves used as fodder, and stems as fuel wood, also used to bath newborn babies to protect them from the different infection. (P.G)Least ConcernGujjar (N = 16)Bakarwal (N = 19)Kashmiri (N = 37)Pahari (N = 31)Dogra (N = 25)0.26128Sapindaceae*Acer caesium* Wall. ex BrandisSMH-402(Ace-cae)Gansu (G)Kaind (K)Kanda (P)Stem LeavesDried stem and fresh leaves are used.Stem and branches are used as fuel wood and leaves as fodder. *Least ConcernGujjar (N = 23)Kashmiri (N = 18)Pahari (N = 24)0.1365*Aesculus indica* (Wall. ex Cambess.) Hook.SMH-110(Aes-ind)Goon (B)Haandoon (K)Khanor (P)Seeds LeavesStem BranchesSeeds are dried and grinded. Raw leaves are used. Stem are used when dry.Seeds are also used to make flour. (P) Leaves are used as fodder. * Stem and branches are used as fuel wood. *Least ConcernBakarwal (N = 27)Kashmiri (N = 21)Pahari (N = 31)0.1679Saxifragaceae*Bergenia ciliata* (Haw.) SternbSMH-422(Ber-cil)Palfut (B)Zakhmehayat (K)Zakheyat (P)Leaves RhizomesLeaves are cooked, sometimes dried. Rhizomes are dried, powdered and used along with water.Leaves are used as food. * Rhizomes are used in treatment of kidney stones. * Leaves are used for making herbal tea. (P)VulnerableBakarwal (N = 45)Kashmiri (N = 31)Pahari (N = 52)0.27128*Bergenia stracheyi* (Hook. f. and Thomson) EnglSMH-423(Ber-str)Sapdotri (G)Katkotar (B)Daindap (K)Mutre (P)Root barkRoot barks are grinded and meshed to obtain an extract.Extracts of root bark are used in treating eye problems. *Least ConcernGujjar (N = 31)Bakarwal (N = 21)Kashmiri (N = 24)Pahari (N = 43)0.24119Scrophulariaceae*Verbascum thapsus* L.SMH-511(Ver-tha)Gamhar (G)Jungli tamook (P)SeedsFlowersRaw seeds are mixed with gee and fried. Fresh flowers are used.Raw seeds are eaten. (G) Flowers are used in herbal tea. *Least ConcernGujjar (N = 41)Pahari (N = 53)0.1994Solanaceae*Datura stramonium* L.SMH-438(Dat-str)Datur (K)Kudle (P,G)Keetr (D)SeedsruitsSun-dried seeds are powdered.Seeds extracts are obtained by boiling seeds in water for more than one hour.Sun-dried seed powder is used to cure cough. (K) Seeds extracts (*Sharbat*) is made along with other herbs like *Rheum webbianum, Artemisia absinthium, Origanum vulgare, Prunella vulgaris, Arnebia benthamii, Viola odorata*, for pregnant mothersSame extracts *(Sharbat*) is given to COVID-19 patients to get relief from chest and throat problems. (K, P, G) Fruits are used to worship Lord Shiva in Hinduism. (D)Least ConcernGujjar (N = 23)Kashmiri (N = 19) Pahari (N = 24)Dogra (N = 32)0.2098*Hyoscyamus niger* L.SMH-455(Hyo-nig)Van tamok (G)Bazarbhang (B)Bhnag (P)SeedsSeeds are sun dried and powdered, used with water.Seeds are used for digestive, diaphoretic, itching and skin disorders. *Least ConcernGujjar (N = 31)Bakarwal (N = 28)Pahari (N = 35)0.1994Taxaceae*Taxus wallichiana* Zucc.SMH-505(Tax-wal)Bririmi (K)Postul (B)BarkBranchesBark is dried and boiled in water for more than 20 min.Branches are used for flooring/bedding huts. *Bark is used to make tea. (B)Critically EndangeredBakarwal (N = 66)Kashmiri (N = 47)0.23113Urticaceae*Urtica dioica* L.SMH-510(Urt-wal)Soi (K)Chichru (P,G)Kandyari (D)SeedsSeeds are dried and powdered and mixed with water and taken orally.Seeds are used as hair tonic and growth stimulants. *Least ConcernGujjar (N = 42)Kashmiri (N = 15)Pahari (N = 37)Dogra (N = 29)0.25123Ulmaceae*Ulmus wallichiana* Planch.SMH-519(Ulm-wal)Bran (K)Peeraval bota (P,G)Stem LeavesDried stems and fresh leaves are used.Stem is used as firewood and leaves are used as fodder. * Stem is also kept in homes to protect from the evil eye. (K)VulnerableGujjar (N = 30)Kashmiri (N = 43)Pahari (N = 17)0.1890Violaceae*Viola odorata* L.SMH-512(Vio-odo)Billar (B)Banafsha (K)Gunafech (D)LeavesCooked.Leaves are used as vegetables. * Least ConcernBakarwal (N = 29)Kashmiri (N = 18)Dogra (N = 42)0.1889(*) is used for all ethnic groups corresponding to the particular plant in the row. Abbreviations: Gujjar (G), Bakarwal (B), Kashmiri (K), Pahari (P), Dogra (D).


### 3.4. Novelty of the Study

Some of the ethnobotanical uses found in our study had never been reported from the region, including the use of the stem of *Betula utilis* to treat diabetes, the roots of *Dolomiaea macrocephala* used for treating body weakness and lower back pain, and the recipe locally called *Nashasta.* Stems of *Eremurus himalaicus* were chewed raw. *Fragaria nubicola, Ulmus wallichiana*, *and Juniperus communis* were used for religious and cultural perspectives. Leaves of *Heracleum candicans* were given to cattle to increase milk production. Flowers of *Trifolium repens* and *Trifolium pratense* were eaten raw. Whole plant of *Thymus linearis* was used by the Gujjar and Pahari to wash and clean the pots used to keep milk. Twigs of *Parrotiopsis jacquemontiana* and *Indigofera heterantha* were used to make traditional fire pots (*Kangri*). A unique food preparation from *Dolomiaea macrocephala* was reported for the first time from the Himalayan region.

### 3.5. Preference Analysis

The preference analysis highlighted the differences in the uses of the plants (χ^2^ = 74.991, df = 8, *p* < 0.001). The local population used plants most commonly as medicine (51.4% responses), followed by food (14.9%), and fodder (9.5%) (Figure 3A). Apart from the need for their own medicinal care, this can also be credited to the high market value of medicinal species when sold to the pharmaceutical industry [53]. It has been widely documented that people in rural areas favor plant-based medicines [44,57,58]. There was very little difference in the species used, nor in their specific use, and all traded species were collected and used all over the region which indicates a very ample distribution of traditional knowledge across many ethnic groups and transgressing national boundaries, as also indicated by Ajaib et al. [58] and Khan et al. [40]. Most plants were frequently used for medicine but we also recorded a high number of species (43) used for food purposes, e.g., *Allium semenovii, Amaranthus blitum, Capsella bursa-pastoris, Dioscorea deltoidea, Dryopteris stewartii, Eremurus himalaicus, Justicia adhatoda, Malva neglecta, Mentha arvensis, Plantago lanceolata,* and *Plantago major* (Table 2). Given the importance of wild edibles for food security, the local people (Gujjar, Bakarwal) strongly believed that wild foods were important to maintain good health, similar to the participants in other areas [59,60]. Wild vegetables, fruits, herbal tea, and spices form an important part of traditional food systems in the region of the Himalayan Mountains [60,61]. A drink (*sharbat*) was given to patients of any age against chest and throat problems and was especially used during COVID-19. *Prunella vulgaris* (Kale voth), *Cichorium intybus, Salix alba* (Yeed), and *Arnebia benthamii* (Kah zaban) were mixed and boiled to make herbal tea. Srivastava et al. [62] reported various plants (*Ocimum sanctum, Nigella sativa, Astragalus membranaceus*) used to help patients with COVID-19. Adhikari et al. [63] documented the use of various plants like *Artemisia annua, Agastache rugosa,* and *Astragalus membranaceus*, for COVID-19. Similarly, Pieroni et al. [64] showed the importance of local plant resources as community responses to COVID-19. 

Different parts of the plants were used, with a significant difference between their uses (χ^2^ = 70.587, df = 9, *p* < 0.001). Leaves were the most highly utilized (25.2%), followed by roots (20.2%), flowers (10.9%), whole plant (9.2%), stem (8.4%), seed oil (7.6%), fruits (5%), bark and branches (4.2% each), rhizomes (3.4%), and twigs (1.7%) (Figure 3B). Based on the differences in plant component preference levels, PCA analysis elucidated three different groups (Figure 4). Roots and leaves were clearly separated from one another, whereas other sections were grouped together. Our findings are similar to the ethnobotanical investigations conducted in other Himalayan regions [65]. Roots were frequently employed because they contained a higher concentration of bioactive components than other plants [66]. The excessive use of roots or whole plants should, however, be discouraged, as this practice can have a negative impact on population and growth [25,67].

### 3.6. Cross-Cultural Usage of Plant Resources

Comparative analysis showed that only fourteen plant species were used by all five ethnic groups (Figure 5A). Plants commonly shared among the different ethnic groups were mainly for medicinal usage (N = 10), followed by fodder (N = 7), food (N = 6), fue wood (N = 4), tea (N = 3), and flooring (N = 1), and religious (N = 1) and spiritual (N = 1) practices. The remarkable heterogeneity on the use of plants could be referred to the different historical stratifications of the studied groups and by the various sociocultural adaptations and the human ecological interactions within the given environments. These types of close affinities of the uses of certain plants among the respective groups could be referred to the fact that some of them underwent some socio-cultural negotiations with others. For instance, Gujjar, Bakarwal, and Pahari are intermarrying, but never intermarry with Dogra because of their different faith and religion. This has in turn impacted the transmission of ethnobotanical knowledge among them. It is also relevant to mention that the large number of use differences found could also be because the ethnic groups occupy very different geographical regions. Bakarwal and Gujjar live at higher elevations, Pahari and Kashmiri occupy middle to higher altitudes, Dogra live in sub-urban environments. It is also important that the Bakarwal are practicing mobile pastoralism, leading to different plant knowledge.

The Gujjar reported the highest number of plants (25%, 94 species), followed by Pahari (24%, 89 species) and Bakarwal (21%, 76 species), while Kashmiri reported (18%, 68 species), and the lowest number of plants was reported by Dogra (12%, 45 species) (Figure 5B). We found a comparatively high overlap in the use of certain plants among ethnic groups. The Gujjar, Bakarwal, and Pahari ethnic groups used 17 plants in common, while all ethnic groups used 14 species in common. One of the reasons for the Gujjar Pahari and Bakarwal overlap might be that both ethnic groups live in highly remote areas in mountainous areas. In contrast the Dogra ethnobotanical portfolio consists of a comparatively small number of plants, as they mostly live in urban and suburban areas and thus have less interaction with nature. Similar cross-cultural plant usage has been recorded from other regions of the Himalayas, e.g., by Abbas et al. [68], who found similar results in Pakistan’s Himalayas and attributed the findings to socio-cultural differences. In the Ladakh region, Haq et al. [4] discovered a notable overlap in plant uses among three ethnic groups (Balti, Beda, and Brokpa). Plants common to all cultures were mostly used for therapeutic purposes, while some were also used for religious purposes in the two major religions (Muslim faith and Buddhism). 

Looking at the plant uses among the different cultural groups, we found that the Dogra, who follow the Hindu religion, had some ritual practices using plants. *Calotropis procera* (aak) had a special religious (Hinduism) with regard to the deity Hanuman in the many regions of Jammu, especially Kathua, Reasi, and Udhampur. Leaves of *Aegle marmelos* (bel-patra), fruits of *Datura stramonium, Ziziphus jujuba* (ber), and seeds of *Elaeocarpus ganitrus* (Rudraksh) were especially attributed to Lord Shiva. Some plants were used in different festivals, for instance oil from the seeds of *Brassica campestris* (Tilgagul) was used in the Deepavali Diwali - the festival of lights. *Cannabis sativa* was used as a drink (bhang) for the Holi festival. In Hinduism, *Azadirachta indica* is not only beneficial to mankind but to the gods as well. *Santalum album* is considered as “wooden gold” in Hinduism due to its fragranced wood. The flowers of *Rhododendron arboreum* (Nichhni) were used to bless crops for never-ending food security. Among Kashmiri, *Ulmus wallichiana* (bran), *Amaranthus blitum* (Craie) and *Celtis australis* (brimji) were also used for religio-magical purposes. Gujjar, Pahari and Kashmiri, bathe the deceased with *Althaea officinalis* (Sazposh) along with “Kafoor” (from *Cinnamomum camphora*). Leaves of *Salix alba* are used to bathe newborns so that they may not catch any infection. Some important medicinal herbs included the seeds of *Datura stramonium* (Datur-bool), *Rheum webbianum* (Pambchalan), *Artemisia absinthium* (Tethwan), *Origanum vulgare* (Wan baber), and *Prunella vulgaris* (Kale voth). *Arnebia benthamii* (Kah zaban) and *Viola odorata* (Banafsha), which were boiled together to make a drink (Sharbat) given to nursing mothers for one month to stimulate the defense mechanism of the body, preventing infections. The sap of fruits from *Fragaria nubicola* was locally used as ink by spiritual healers to write scrolls and amulets. *Celtis australis, Datura stramonium* had spiritual and cultural values [69,70,71]. In Kashmir, *Ulmus wallichiana* (bran), *Celtis australis* (brimji), and *Amaranthus blitum* (Craie) were used to protect children from the evil eye and black magic. Similar uses were reported by Haq et al. [4] in Ladakh, Trans-Himalayas, and Rout et al. [72] in Assam (northeast India). 

### 3.7. Quantitative Ethnobiological Approach 

The two-way cluster analyses of eight ethnobotanical uses of the 127 species resulted in 6 major clusters (Figure 6) between provisioning services and plant species that were also recognized based on Sorenson’s similarity index. *Abies pindrow* and *Picea smithiana* were both used for timber and flooring and formed the first group of the cluster, whereas plants such as *Celtis australis* and *Parrotiopsis jacquemontiana* used for agricultural equipment, formed the second cluster; and plants like *Betula utilis* and *Woodfordia fruticosa* used for making herbal tea, formed the third cluster. Plants like *Aesculus indica, Indigofera heterantha, Salix alba, Acer caesium, Ziziphus jujube* and *Prunus cornuta* used for both fuelwood and fodder, formed the fourth cluster. Species like *Arnebia benthamii, Eremurus himalaicus, Justicia adhatoda, Fragaria nubicola, Polygonum amplexicaule* and *Plantago lanceolata* were used for food and medicinal purposes, and forming the 5th and 6th clusters of the dendrogram, respectively. The clustering of plants based on ethnobotanical usage is presented in Figure 7 where plants grouped into one limb are more similar in usage and show proximity to each other. This was further substantiated by PCA analysis, which revealed separate groups of provisioning services based on differences in plant utilization preference levels (Figure 7). Medicinal and food services were split, while other provisioning services were divided into various categories. Upadhyay et al. [73] observed similar results when using PCA to group plant consumption by local communities, finding 59% variation in their data. In the Lesser Himalayas of Pakistan, Rahman et al. [74] employed a two-way cluster and principal component analysis to quantitatively classify different diseases. The PCA found that all plant species were positively connected. Asif et al. [2], from the Indian Himalayan region, employed quantitative categorization factors to report five groupings of wild plants from tribal populations in the tehsil Karnah within Jammu and Kashmir, India. Haq et al. [75], Tokuoka et al. [76], Dicko et al. [77], and Ahoyo et al. [78] applied multivariate analysis for quantitative ethnobiological research.

### 3.8. Conservation of Key Plant Species

Based on the conservation assessment, out of 127 species observed, 14(11%) species fell in the Critically Endangered category of IUCN, 8 (6%) were Vulnerable, 2(2%) were Endangered, and 103 (81%) in the Least Concern category (Table 2). Many important medicinal plants such as *Aconitum heterophyllum, Trillium govanianum, Rheum webbianum, Fritillaria cirrhosa, Aucklandia costus, Bergenia ciliata, Dolomiaea macrocephala, Dactylorhiza hatagirea, Inula racemosa*, and *Picrorhiza kurroa* are facing threats because of the exhaustive utilization of their underground parts on a large scale. Due to the over-collection of these species for trade purposes they have become en-dangered all over the Himalayan region [79,80], with all medical systems of South Asia utilizing Himalayan plants in their preparations [81]. With the expansion of roads and the growing popu-lation, plants have now been exposed to new commercial interests [82]. It is estimated that over 15000 plants used in the traditional healthcare systems in the world fall in the threatened cate-gory and need immediate conservation efforts to prevent their extinction [83]. Local people rely on plant species for medicine, food, and for their cultural and religious value. The above-mentioned species have become a favourite target due to their use by indigenous communities in traditional medicine systems and other uses, resulting in species decline. This unsustainable exploitation endangers biodiversity and the livelihoods of indigenous peoples. It is critical to organize awareness programmes and conservation education involving stakeholders and ethnic people in order to prevent changes in the population dynamics of these species. Traditional knowledge bearers and their knowledge can assist to promote the protection of species and habitats more effectively, stimulate the sustainable use of biodiversity, as well as raise awareness of the importance of conservation. Better understanding of the specific knowledge of medicinal plants, for example, could aid in the development of more complex community-based conservation efforts. Local knowledge, as well as area economic and socio-cultural issues, can be considered in inclusive conservation initiatives (e.g., perceptions based on local values and beliefs). Improved recognition of local knowledge may also aid in the preservation and transmission of local knowledge, which is required for the continuation of local—often still sustainable—land-use practices. It is important to note that when developing conservation strategies, local traditions and customs must be considered.

## 4. Conclusions

The present study found that religious affiliation can be an important factor influencing the use of wild plants, as it shapes kinship relations and vertical transmission of traditional local ethno-botanical knowledge. This study emphasizes the importance of ethnobotanical uses of wild plants and underlines that the traditional knowledge associated with their use is vanishing. The local population in the study area was often not fully aware of the value and need for conservation of their plant resources (conservation, religious, ecological, and aesthetic values) and exploiting them at an increasing rate. The depletion of indigenous knowledge is already accelerating due to the disinterest of the young generation to obtain such knowledge. The interruption of oral-based knowledge transfer, the extinction of valuable medicinal species due to overharvesting and other anthropogenic factors, and the influence of the modern allopathic system of medicines can be considered the major factors leading to knowledge decline. The documentation of traditional knowledge serves a wide variety of purposes, including its preservation for future generations, it is safeguarding by putting the information in the public domain, and its use as the baseline for further research and conservation strategies.

## Figures and Tables

**Figure 1 biology-11-00491-f001:**
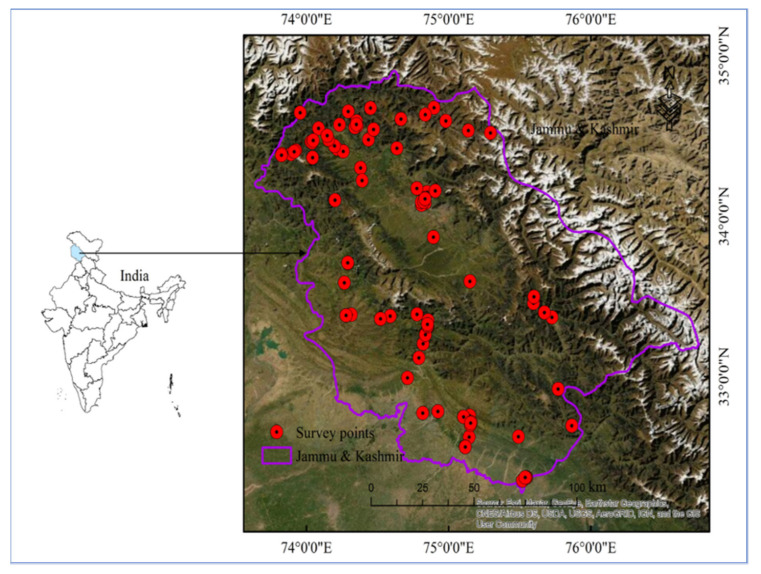
Map of the Jammu and Kashmir, India and points showing the survey villages in Jammu and Kashmir western Himalayan region, India.

**Figure 2 biology-11-00491-f002:**
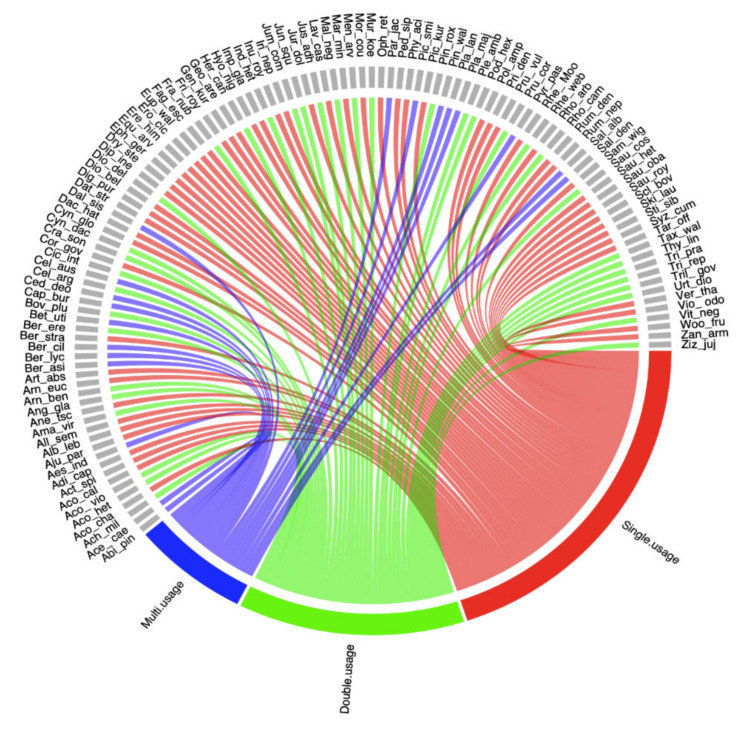
A Chord diagram depicting plant species distribution allied to three usage patterns in the Western-Himalayan region of Jammu and Kashmir, India. The full name of each plant is provided in Table 2. The thickness of each bar reflects the degree of variation in each use category, and the direction of the lines depicts which plant is associated with which form of usage.

**Figure 3 biology-11-00491-f003:**
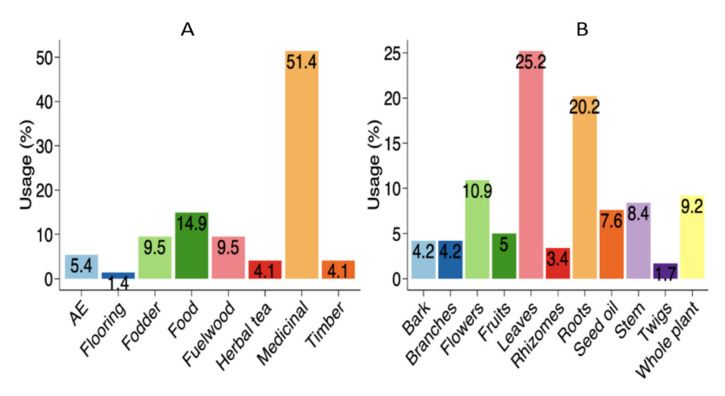
(**A**) Percentage distribution of plant species in various major use categories, and (**B**), Percentage of plant parts used in Jammu and Kashmir Western Himalayan region, India. AE: Agricultural equipment.

**Figure 4 biology-11-00491-f004:**
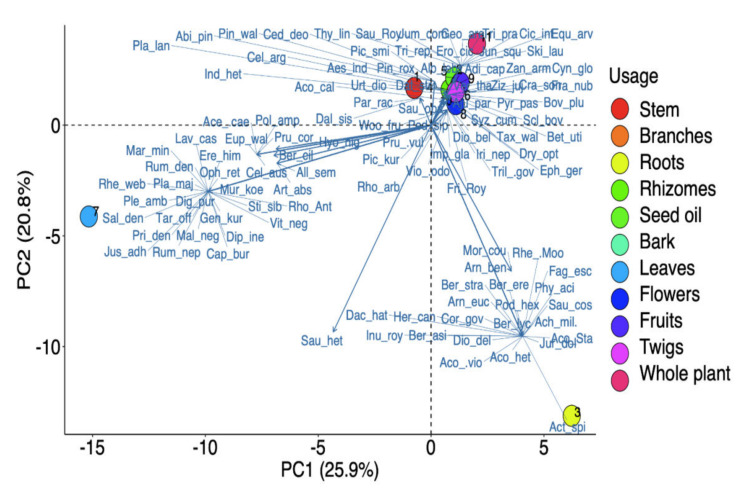
Principal component analyses (PCA) biplot of different plant part usage investigated in Jammu and Kashmir Western Himalayan region, India.

**Figure 5 biology-11-00491-f005:**
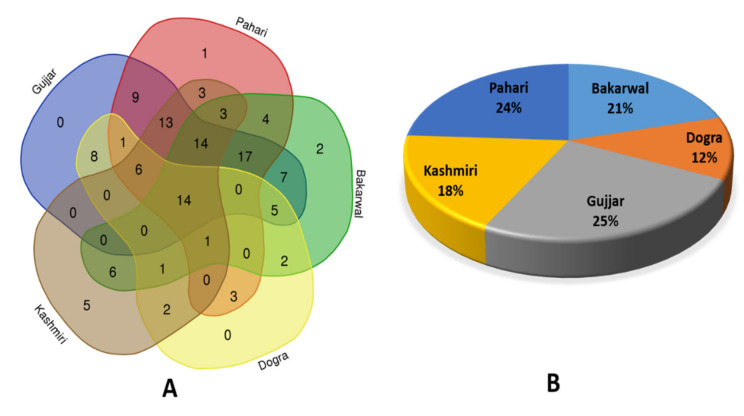
(**A**) Plant usage patterns, and (**B**) percentage use of plants across different ethnic groups, in Jammu and Kashmir Western Himalayan region, India.

**Figure 6 biology-11-00491-f006:**
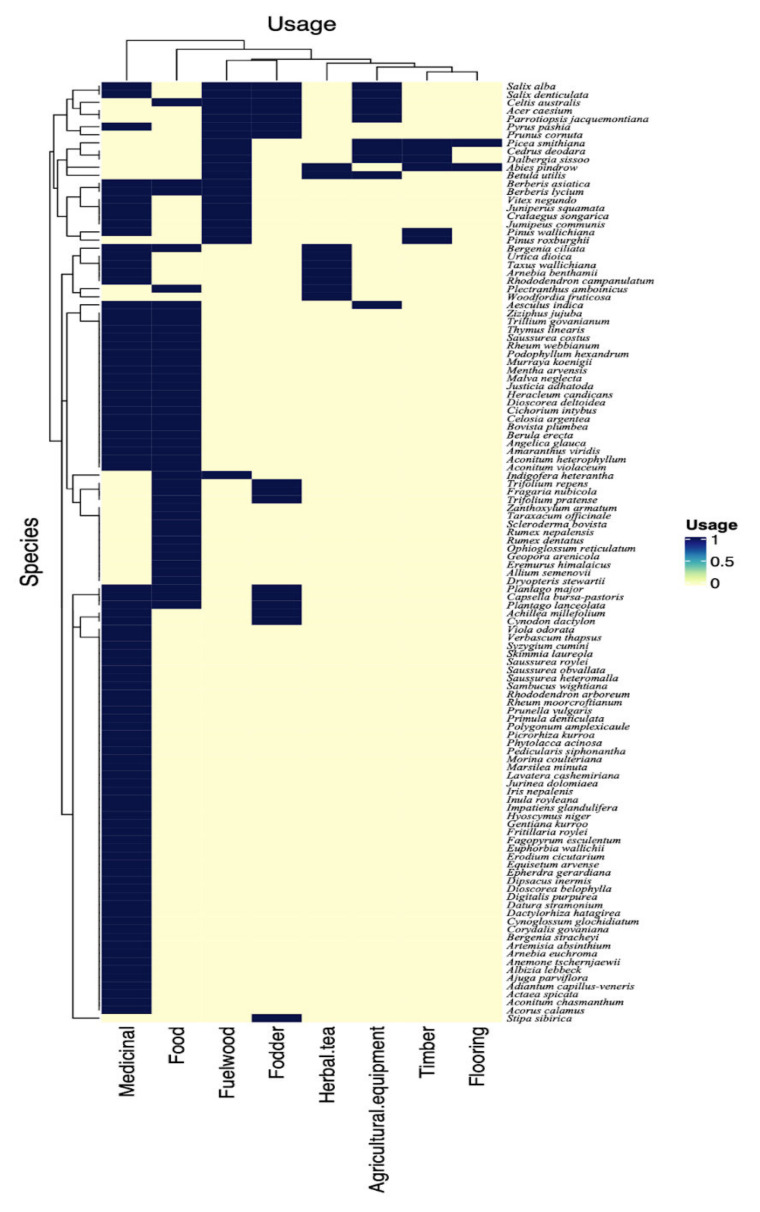
Two-way cluster analysis based on Sorenson’s similarity index between provisioning services and plant species in Jammu and Kashmir Western Himalayan region, India.

**Figure 7 biology-11-00491-f007:**
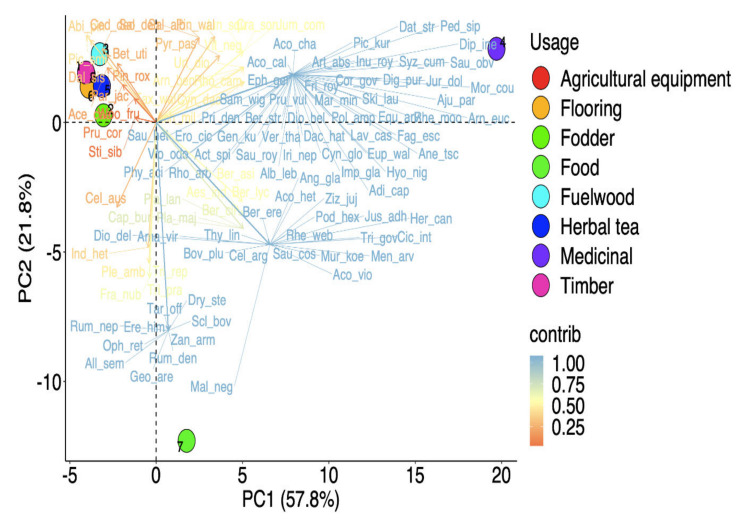
Principal component analyses (PCA) biplot of different provisioning services investigated in Jammu and Kashmir Western Himalayan region, India.

## Data Availability

Data are available from the first author.

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
