# Peer review of "A Cross-Cultural Analysis of Plant Resources among Five Ethnic Groups in the Western Himalayan Region of Jammu and Kashmir"

_biology, 2022, doi:10.3390/biology11040491_

Round 1
Reviewer 1 Report
A solid , well designed and interesting paper. Definitely of interest for the readers. I really have no negative comments on it. Maybe that it looks more like a book chapter than a paper. But I strongly suggest the direct acceptance, considering the complex data, the lack of well designed studies from this area within and its increase significance in this area of research.
Author Response
Response to Reviewers 1
Comments
A solid, well designed and interesting paper. Definitely of interest for the readers. I really have no negative comments on it. Maybe that it looks more like a book chapter than a paper. But I strongly suggest the direct acceptance, considering the complex data, the lack of well-designed studies from this area within and its increase significance in this area of research.
Response
Thanks for your encouraging comments.
Reviewer 2 Report
Dear author (s),
I had the pleasure to review your manuscript, which I've found interesting, but I note some gaps that need to be addressed. Some considerations for revision are given:
Study Area and Communities: Ethnobotanical studies require a more detailed description of the study area. For example: the total area of the study, altitude, population (males and females), plants that contribute to the income of these communities, among others.
Informant selection was based on the Snowball Sampling technique. However, what were the criteria for selecting the key-person? What would be the reason for having more male than female respondents? (especially in Kashmiri). It would be interesting to have more information about it.
There are errors in the numbering of figures and tables.
Table 7 is presented after Table 8. Therefore, the discussion of each figure is wrong (Quantitative Ethnobiological Approach section).
Title of figure 6B – What percentage do you mean?
It is required to improve the quality of Figure 7.
Conservation of Key Plant Species: this section seems underdeveloped. With the information you have you could discuss your results in another way, in accordance with the third objective.
There are some typing errors across the manuscript (l 130, 305, etc.)
Cite web pages in text as you would any other source, using the journal citation style (Line 140, 159, 186).
I don't feel qualified to judge about the English language and style
Author Response
Response to Reviewers 2.
Dear author (s),
I had the pleasure to review your manuscript, which I've found interesting, but I note some gaps that need to be addressed. Some considerations for revision are given:
Comment
Study Area and Communities: Ethnobotanical studies require a more detailed description of the study area. For example: the total area of the study, altitude, population (males and females), plants that contribute to the income of these communities, among others.
Response
Modified as suggested. Line no. 135 to 139; 141 to 144; 150 to 152
Comment
Informant selection was based on the Snowball Sampling technique. However, what were the criteria for selecting the key-person? What would be the reason for having more male than female respondents? (especially in Kashmiri). It would be interesting to have more information about it.
Response:The first participants were local people who were indicated by local and village authorities as knowledgeable in plant use. They then indicated further possible participants and so on. The reason that our study had less women participating lies in the social, religious and cultural values of the society. Women, on the whole, are much more restricted to the household, and often not allowed or not willing to interact freely with strangers.
Line No. 164 to 167
Comment
There are errors in the numbering of figures and tables.
Table 7 is presented after Table 8. Therefore, the discussion of each figure is wrong (Quantitative Ethnobiological Approach section).
Response
Now placed as per the serial number.
Comment
Title of figure 6B – What percentage do you mean?
Response
The percentageuse of plants across different ethnic groups in Jammu and Kashmir Western-Himalayan region, India
Comment
It is required to improve the quality of Figure 7.
Response
Modified as suggested
Comment
Conservation of Key Plant Species: this section seems underdeveloped. With the information you have you could discuss your results in another way, in accordance with the third objective.
Response
Modified according the suggestion Line no. 464 to 470; 445 to 479
Comment
There are some typing errors across the manuscript (l 130, 305, etc.)
Response
Entire manuscript has been thoroughly checked and corrected accordingly.
Comment
Cite web pages in text as you would any other source, using the journal citation style (Line 140, 159, 186).
Response
Changed as per suggestions.
Comment
I don't feel qualified to judge about the English language and style
Response
The entire Manuscript has been thoroughly checked again and corrected where errors were found

Reviewer 3 Report
A well designed and written paper. The authors have been thorough in their research. A few typos that need correcting. Highlighted in the ms.

Author Response
Response to Reviewers 3.
Comment
A well designed and written paper. The authors have been thorough in their research. A few typos that need correcting. Highlighted in the ms.
Response
Thanks for your encouraging comments. The entire Ms, is checked again to correct typos errors.
Round 2
Reviewer 2 Report
The authors have satisfactorily addressed my comments and I may recommend the manuscript for publication.